# Chirality selective magnon-phonon hybridization and magnon-induced chiral phonons in a layered zigzag antiferromagnet

Jun Cui[1], Emil Viñas Boström[2], Mykhaylo Ozerov [3] ✉, Fangliang Wu[1], Qianni Jiang[4], Jiun-Haw Chu [4], Changcun Li[5], Fucai Liu [5], Xiaodong Xu [4,6], Angel Rubio [2,7] ✉ & Qi Zhang [1] ✉

Two-dimensional (2D) magnetic systems possess versatile magnetic order and can host tunable magnons carrying spin angular momenta. Recent advances show angular momentum can also be carried by lattice vibrations in the form of chiral phonons. However, the interplay between magnons and chiral phonons as well as the details of chiral phonon formation in a magnetic system are yet to be explored. Here, we report the observation of magnon-induced chiral phonons and chirality selective magnon-phonon hybridization in a layered zigzag antiferromagnet (AFM) FePSe$_3$. With a combination of magneto-infrared and magneto-Raman spectroscopy, we observe chiral magnon polarons (chiMP), the new hybridized quasiparticles, at zero magnetic field. The hybridization gap reaches 0.25 meV and survives down to the quadrilayer limit. Via first principle calculations, we uncover a coherent coupling between AFM magnons and chiral phonons with parallel angular momenta, which arises from the underlying phonon and space group symmetries. This coupling lifts the chiral phonon degeneracy and gives rise to an unusual Raman circular polarization of the chiMP branches. The observation of coherent chiral spin-lattice excitations at zero magnetic field paves the way for angular momentum-based hybrid phononic and magnonic devices.

Strongly coupled spin and lattice degrees of freedom play essential roles in diverse quantum materials phenomena, from driving multiferroic polarization to manipulating magnetic order via lattice vibrations[1,2]. Collective spin and lattice excitations with finite angular momenta, e.g., magnons and chiral phonons, are of particular importance in processes related to angular momentum transfer and conservation, e.g., in spintronics and ultrafast magnetism[3]. The strong coupling between magnons and chiral phonons is expected to offer a coherent path for energy and angular momentum transfer between the two degrees of freedom, and form a hybrid spin-lattice excitation with chirality.

As a dynamical form of coherent spin-lattice interactions, the magnon-phonon strong coupling has received great theoretical attention[4–9], where coupling-induced topological magnons have been predicted for 2D magnetic systems[6–9]. Experimentally, the characteristic avoided crossing between the magnon and phonon bands has

[1]National Laboratory of Solid State Microstructures and Department of Physics, Nanjing University, 210093 Nanjing, China. [2]Max Planck Institute for the Structure and Dynamics of Matter, Luruper Chaussee 149, 22761 Hamburg, Germany. [3]National High Magnetic Field Laboratory, Florida State University, Tallahassee, FL 32310, USA. [4]Department of Physics, University of Washington, Seattle, WA 98195, USA. [5]School of Optoelectronic Science and Engineering, University of Electronic Science and Technology of China, 611731 Chengdu, China. [6]Department of Materials Science and Engineering, University of Washington, Seattle, WA 98195, USA. [7]Center for Computational Quantum Physics, The Flatiron Institute, New York, NY 10010, USA. ✉e-mail: ozerov@magnet.fsu.edu; angel.rubio@mpsd.mpg.de; zhangqi@nju.edu.cn

been observed in both 3D[10–13] and layered magnets[14–19] indicating the strong coupling and the formation of the hybrid quasiparticles, magnon polarons (MP). However, the strong coupling of magnons and chiral phonons remains elusive, also the details and prerequisites of chiral phonon formation in a magnetic system are largely unclear.

Recently discovered van der Waals (vdWs) magnets[20–25] exhibit strong spin-lattice interactions and possess unprecedented tunability via pressure[26], strain[27], electrostatic gating[28–30], and moiré engineering[31–33], thereby offering an ideal platform to explore chiral spin-lattice excitations in the 2D limit. A large class of vdWs magnets crystallizes with a honeycomb spin lattice, where the magnetic order ranges from simple patterns such as ferromagnetic and Néel-type order to the more complex orders of stripe and zigzag AFM[34]. Crucially, the magnon symmetry differs depending on the magnetic ground state and the space group symmetries. On the lattice side, honeycomb lattices are known to support chiral phonons at the $K$ and $K'$ points as well as degenerate chiral phonons at the zone center[35,36]. Therefore, 2D honeycomb magnets are expected to host versatile and highly tunable magnon- (chiral) phonon interactions and hybrid quasiparticles, which are yet to be demonstrated. Due to the high frequency and low dissipation of AFM excitations[37,38], strongly coupled chiral phonons and AFM magnons are of particular interest.

In this work, we report the observation of magnon-induced chiral phonons and the formation of chiral magnon polarons (chiMP) in the zigzag AFM insulator FePSe₃. We first demonstrate that the zigzag magnetic order survives down to the atomic limit in FePSe₃ via optical linear dichroism (LD) measurements. Combining magneto-infrared and magneto-Raman spectroscopy, we unveil an unusual avoided-crossing pattern suggesting a selective strong coupling between a pair of degenerate AFM magnons and a pair of nearly degenerate optical phonons. A clear hybridization gap is observed at zero magnetic field with the coupling strength reaching 0.25 meV (60 GHz), which is 1.8% of the phonon frequency. The strong-coupling features remain visible down to quadrilayers. By combining polarization-resolved Raman measurement and first-principle calculations, we reveal that the pair of phonons involved in the coupling are chiral and carry opposite angular momenta. Each chiral phonon mode selectively hybridizes with the AFM magnon of parallel angular momentum constrained by symmetry. This selective coupling well explains the observed chiMP dispersions and their anomalous Raman circular polarization. Remarkably, the chiral phonon degeneracy remains lifted even at very large magnon-phonon detunings, reaching 10% of the resonance frequency. Our findings not only unveil novel chiral spin-lattice quasiparticles, but also provide a path forward to utilizing them for angular momentum-encoded information processing in novel spin-phononic devices.

## Results and discussion

The vdWs antiferromagnetic insulator FePSe₃ belongs to the class of transition metal phosphorous trichalcogenides ($MPX_3$, $M$ = Fe, Mn, Ni, and $X$ = S, Se). The Ising-type magnetic moments are mainly carried by the Fe atoms, arranged in a 2D hexagonal lattice, and are oriented out-of-plane. In the ground state, FePSe₃ displays a zigzag AFM order[39,40], as schematically shown in Fig. 1a. The formation of zigzag spin chains breaks the three-fold rotational symmetry, and yields an in-plane optical anisotropy that can be detected via optical linear dichroism (LD) measurements in the reflection geometry[41] (see "Methods" section). It has been observed in other zigzag AFMs within the $MPX_3$ family[41–43]. The LD signal, defined as LD = $(I_{\parallel} - I_{\perp})/(I_{\parallel} + I_{\perp})$, is proportional to the magnitude square of the AFM order parameter, where $I_{\parallel}$ and $I_{\perp}$ stand for the reflected light intensity along and across the zigzag direction, respectively. Figure 1b shows the typical temperature dependence of the LD in a few-layer FePSe₃ flake. As the temperature increases, the LD signal gradually decreases and finally vanishes above the Néel temperature ($T_N$ ~ 108 K). As for atomically thin FePSe₃ flakes, thanks to the strong out-of-plane single-ion anisotropy, the zigzag AFM order survives at least down to the bilayer limit, where an LD of 0.1% is detected as shown in Fig. 1c. The LD from monolayers is less than $10^{-4}$, which is below our instrument sensitivity.

Before discussing the observation of magnon-phonon hybridization in FePSe₃, we first investigate how the magnon polarons resulting from different types of magnon-phonon coupling in an antiferromagnet would behave under external magnetic fields. Consider a pair of degenerate AFM magnons ($\omega_{mag}$) and a pair of degenerate phonons ($\omega_{ph}$) whose frequencies coincide ($\omega_{mag} = \omega_{ph}$). In the case of no interaction, as illustrated in the left panel of Fig. 2a, an external magnetic field along the spin direction lifts the degeneracy of AFM magnons leading to a magnon Zeeman splitting with a $g$-factor of 2. No hybridization of magnons and phonons can be found at the crossing point (0 T). The middle panel of Fig. 2a presents a second scenario, where the magnons and phonons couple non-selectively, as detailed in the phenomenological coupling matrix. In this case, hybridization gaps appear and three MP modes are visible at zero field, with the middle MP branch being doubly degenerate. The right panel of Fig. 2a shows a third scenario, where the magnons and phonons couple selectively. More precisely, each phonon mode only interacts with one of the magnon modes, and vice versa. In this case, there are two MP bands at 0 T, and once a field is applied, they further split into four MP branches. The outer two branches are more magnon-like and exhibit linear Zeeman splitting at high magnetic fields, while the inner ones are more phonon-like and approach each other asymptotically as the field increases.

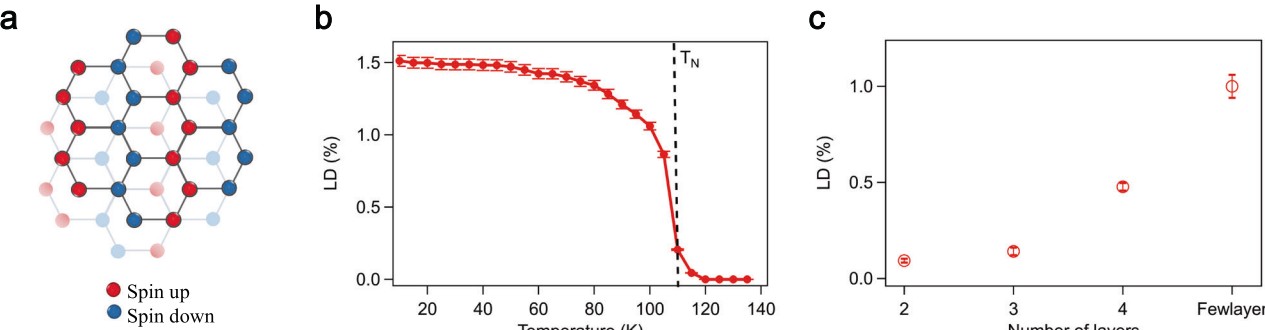

**Fig. 1 | The zigzag spin order and optical linear dichroism of FePSe₃. a** Lattice and spin structure of FePSe₃. It has a zigzag AFM order, where the nearby zigzag chains have an opposite spin orientation (indicated by the blue and red dots), pointing out-of-plane, with rhombohedral interlayer stacking. **b** Optical linear dichroism (LD) as a function of temperature. The LD is proportional to the magnitude squared of the AFM order parameter. It vanishes above the Néel temperature (108 K). **c** LD as a function of layer number in few-layer FePSe₃ flakes. The zigzag AFM order survives down to the bilayer flakes. Error bars represent one standard deviation throughout the work.

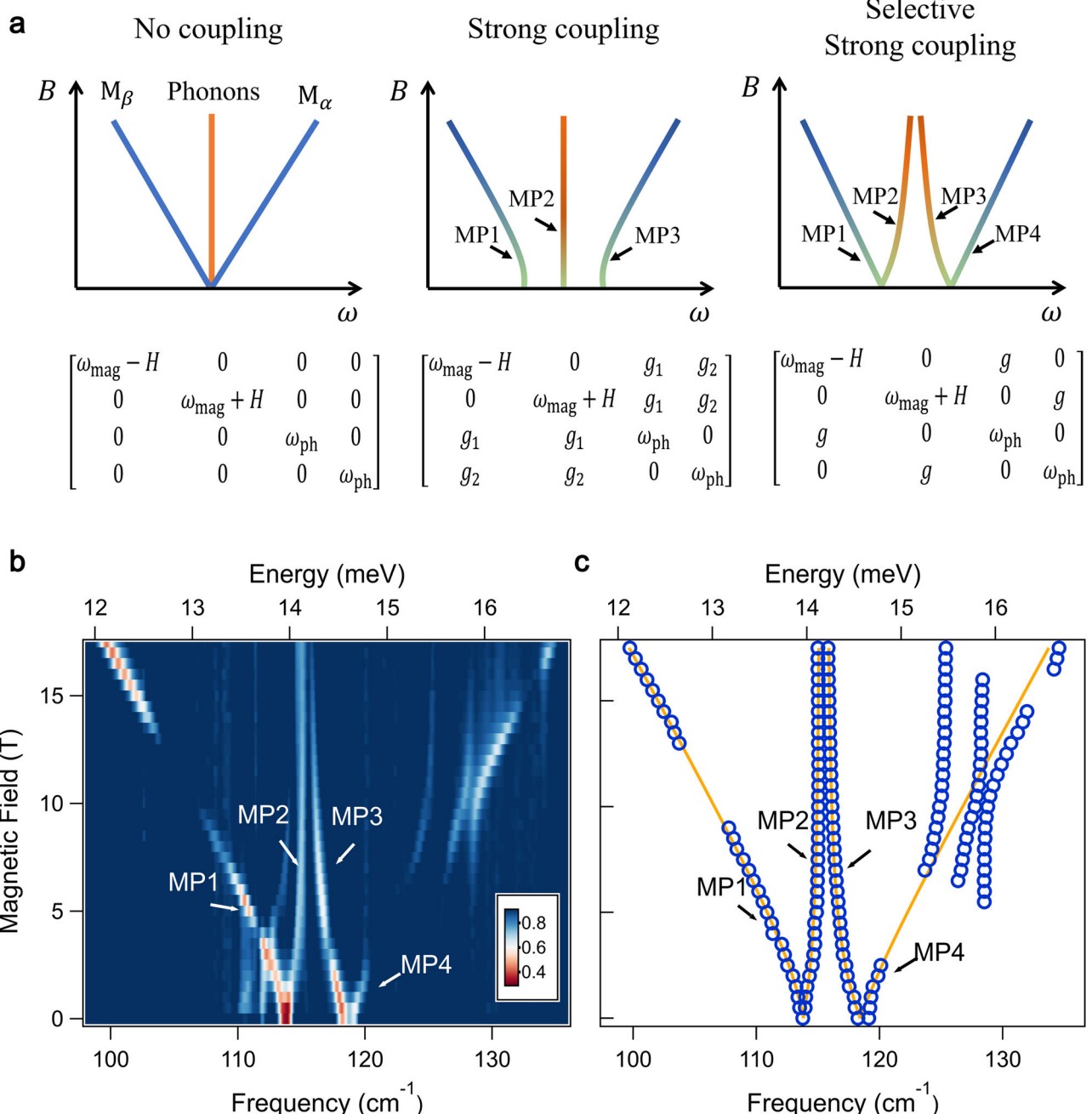

**Fig. 2 | Selective strong coupling of magnons and phonons in FePSe₃.**
**a** Schematic diagrams of magnon polaron (MP) bands under different types of coupling and their phenomenological coupling matrixes. A pair of degenerate AFM magnons ($M_\alpha$ and $M_\beta$) and a pair of degenerate phonons with identical frequencies are considered ($\omega_{mag} = \omega_{ph}$). The magnon Zeeman energy is $H$. Left panel: no interaction between magnons and phonons. Middle panel: non-selective strong coupling, where all magnon and phonon modes couple identically. Right panel:

selective strong coupling, where each phonon mode only couples with one magnon mode. **b** Normalized far-infrared transmission spectra of bulk FePSe₃ up to 17 T, four MP branches are identified (MP1 to MP4). **c** Peak positions of MP branches (blue circles) fitted with selective strong coupling model (orange lines). The deviation of the MP4 fitting line from the experimental peak positions is due to a series of additional magnon-phonon avoided crossings. Above 20 T, they are in good agreement (see Supplementary Fig. S3 for spectra up to 35 T).

We now demonstrate the experimental observation of magnon-phonon strong coupling on a FePSe₃ bulk flake (~50 μm in thickness). We performed magneto-infrared transmission measurements up to 17 T at the Maglab (see "Methods" section). The magnetic field is out-of-plane and parallel to the spin direction (Faraday geometry). The measurement was performed at 4.2 K, well below the antiferromagnetic transition temperature. Normalized far-infrared transmission spectra as a function of magnetic field are presented in Fig. 2b (see Supplementary Fig. S2 for raw data), and associated peak positions are shown in Fig. 2c. At 0 T, two prominent modes can be seen at

113 cm⁻¹ and 118 cm⁻¹, and further split into four bands (MP1 to MP4) for finite magnetic fields. These modes are the main focus of the following discussion. The vanishing signals of MP1 between 105 and 108 cm⁻¹ and MP4 between 120 and 124 cm⁻¹ are due to the strong reflection of infrared (IR) active phonons. At high magnetic fields, MP1 and MP4 shift linearly with a slope of 0.93 cm⁻¹T⁻¹, corresponding to the Zeeman shift of quasiparticles with a magnetic moment of $2\mu_B$, a signature of AFM magnons in FePSe₃. The branches MP2 and MP3 are more phonon-like at high magnetic fields, and barely shift above 10 T. By diagonalizing the coupling matrix, we extract the zero-field AFM

magnon frequency to be $116.8\,\mathrm{cm^{-1}}$ (see Supplementary text S1), which matches our linear spin wave calculation (Supplementary text S4). The pair of phonons are nearly degenerate as listed in Table 1. A magnon-phonon coupling strength of $2.1\,\mathrm{cm^{-1}}$ is obtained for both magnon-phonon pairs, which is 1.8% of the resonance frequency. The deviation of the fit of MP4 to the experimental peak positions between 125 and $135\,\mathrm{cm^{-1}}$ is the result of a series of additional magnon-phonon avoided crossings[16]. At higher fields, the fit agrees well with the observation (see Supplementary Fig. S3 for the spectra up to 35 T). All of these experimental features are consistent with the selective strong coupling scenario mentioned above. The data thus suggests that the pair of AFM magnons in FePSe$_3$ not only coincide in energy with a pair of nearly degenerate optical phonons, but also exhibit selective hybridization with them.

The AFM magnons in FePSe$_3$ are IR active via magnetic dipole transitions, while the phonons involved in the strong coupling are Raman active and are brightened in the far-infrared spectra due to the magnon-phonon coupling[16]. To better resolve the phononic modes involved in the selective hybridization, we turned to polarization-resolved magneto-Raman spectroscopy. The Raman scattering signals of magnon polarons are mainly from the cross-circular channels. For instance, the $\sigma_R\sigma_L$ channel represents the Raman spectrum with right-handed ($\sigma_R$) light excitation and left-handed ($\sigma_L$) light detection. Figure 3a shows Raman spectra of a 96 nm-thick FePSe$_3$ flake as a function of temperature at zero magnetic field. At 4 K, two MP peaks are found at $113.2\,\mathrm{cm^{-1}}$ and $117.1\,\mathrm{cm^{-1}}$ and match with far-infrared results. As the temperature increases, the low-energy MP mode red-shifts and broadens, and the high energy MP mode shifts to $115\,\mathrm{cm^{-1}}$. Meanwhile, the Raman intensity redistributes among the two MP modes. At 70 K, only one peak at $115\,\mathrm{cm^{-1}}$ remains visible and its Raman intensity equals the sum of the two MP modes at 4 K. All these features can be understood by considering the magnon softening with increasing temperature. The red dots are extracted uncoupled AFM magnon frequencies assuming a temperature-independent coupling strength $g$. Above 70 K, the magnons and phonons barely hybridize due to the large detuning. When the magnon-phonon detuning shrinks with decreasing temperature, the hybridization gap opens and pushes away the two MP modes, and the Raman intensities of the two MPs redistribute accordingly.

Figure 3 b presents the Raman spectra of the MP branches as a function of magnetic field in one of the cross-circular channels ($\sigma_R\sigma_L$) (see Supplementary Fig. S5 for details). The corresponding peak positions are shown in Fig. 3c. Two MP modes appear at zero field and split into four MP branches, which is consistent with the selective strong coupling picture and also matches with the far-infrared results. The Raman processes in the $\sigma_R\sigma_L$ and $\sigma_L\sigma_R$ channels have opposite angular momenta transfer from photons to quasiparticles. Hence, to unveil the chirality of the MP branches, we measure the degree of circular polarization (DCP) of Raman spectra, which is defined as $(\sigma_R\sigma_L - \sigma_L\sigma_R)/(\sigma_R\sigma_L + \sigma_L\sigma_R)$. The magnetic field dependence of DCP is shown in Fig. 3d. First, all four MP branches exhibit finite circular polarization suggesting they may carry finite angular momenta. Of particular interest are the phonon-like MP2 and MP3 branches. Remarkably, they show considerable DCP even at 9 T, where the magnon-phonon detuning reaches 10% of the resonance frequency. In addition, the two magnon-like branches MP1 and MP4 exhibit the same DCP under given magnetic field directions, which is drastically different from the regular AFM magnon Zeeman splitting, where two magnons have opposite circular polarization as observed in FePS$_3$ (see Supplementary Fig. S7). This anomalous Raman polarization will be explained in the latter part of the paper.

To further explore the selective magnon-phonon hybridization in the atomic limit, we perform magneto-Raman microscopy on few-layer FePSe$_3$ flakes. Layer-dependent Raman spectra are shown in Fig. 4a. The two MP peaks are observed in the quadrilayer (4L) flakes, are barely seen in trilayer (3L) ones at zero magnetic field. As the layer number reduces, the magnon frequency red shifts due to the weakened AFM order. It enlarges the magnon-phonon detuning and results in an imbalanced Raman intensity among the two MP modes. The Raman mapping of a representative FePSe$_3$ few-layer flake is shown in Fig. 4b, where the color represents the integrated intensity of the low-energy MP mode (indicated by the red-arrow in Fig. 4a). The inset figure shows the corresponding optical image of the flake. The magnetic field dependence of the MP modes in 4L flakes is presented in Fig. 4c. The field-induced shift of all four MP modes can be resolved as marked by the arrows. The corresponding peak positions can be well described with the selective strong coupling model, as shown in Fig. 4d. The extracted uncoupled magnon frequency of 4L flakes is $113.2\,\mathrm{cm^{-1}}$, exhibiting a slight redshift to its bulk value, while the magnon-phonon coupling strength remains strong reaching $2.0\,\mathrm{cm^{-1}}$, as listed in Table 1.

The above experimental results establish the selective strong coupling of magnons and phonons in both bulk and atomically thin FePSe$_3$. Microscopically, questions regarding the nature of the phonons and the selective hybridization remain. To address these questions, we performed first principle calculations of the magnetic and mechanical structure of the material (see Methods). The calculations identify two Raman active phonon modes (P$_1$ and P$_2$), whose frequencies are close to the AFM magnons (see Table 1). These modes belong to the $A_g$ and $B_g$ representations of the point group $C_{2h}$ of the zigzag state, but originate from the $E_g$ representation of the $D_{3d}$ point group of the paramagnetic state. The calculated frequency difference between the P$_1$ and P$_2$ modes is $2.6\,\mathrm{cm^{-1}}$, and arises due to the symmetry breaking associated with a small structural distortion at the magnetic phase transition. The measured value is $<0.1\,\mathrm{cm^{-1}}$, which is only 0.08% of the phonon frequency and 5% of the magnon-phonon coupling strength, making the two phonons nearly degenerate. The larger theoretical value is likely due to the approximation of the exchange-correlation energy introduced in the calculation.

The atomic displacement patterns of the Fe atoms in the P$_1$ and P$_2$ modes are schematically shown in the left panel of Fig. 5a. Crucially, the Fe atoms exhibit out-of-phase motions between nearest neighbors both inside and between the zigzag chains, and along two orthogonal directions. Therefore, the coherent superposition of P$_1$ and P$_2$ with a $\pm\pi/2$ phase shift leads to a pair of circular phonon modes P$_\pm$, as shown in Fig. 5a. Our calculations show that the P$_\pm$ phonons are chiral and carry opposite non-zero angular momenta given by $L^z_\pm = \mp\hbar|\boldsymbol{\epsilon}_1 \pm i\boldsymbol{\epsilon}_2|^2$, where $\boldsymbol{\epsilon}_1$ and $\boldsymbol{\epsilon}_2$ are the polarization vectors of the P$_1$ and P$_2$ modes, respectively, on an arbitrary Fe atom (see Supplementary Text S4–7). This result follows from the definition of the phonon angular momentum, $\mathbf{L} = \sum_{i\alpha} M_{i\alpha}\mathbf{u}_{i\alpha} \times \dot{\mathbf{u}}_{i\alpha}$[44], where $M_{i\alpha}$ and $\mathbf{u}_{i\alpha}$ are the

### Table 1 | Magnon-phonon strong coupling parameters of FePSe$_3$

| (cm$^{-1}$) | IR (50 μm) | | Raman (96 nm) | | Raman(4L) | | Theory | |
|---|---|---|---|---|---|---|---|---|
| | ω | g | ω | g | ω | g | ω | g |
| Magnon (0 T) | 116.8 | – | 115.4 | – | 113.2 | – | 111.8 | – |
| Phonon (P$_1$) | 115.3 | 2.1 | 115.2 | 2.1 | 114.5 | 2.0 | 109.5 | 2.16 |
| Phonon (P$_2$) | 115.4 | 2.1 | 115.2 | 2.1 | 114.5 | 2.0 | 112.1 | 2.0 |

The mode frequencies listed in the table are for uncoupled magnons and phonons. Experimental values of the coupling parameters are extracted via diagonalization of the coupling matrix (see Supplementary Text S1). The theoretical values are from first-principle calculations. As the reduction of the sample thickness, the AFM magnon exhibits a slight red shift. The measured P$_1$ and P$_2$ phonons are nearly degenerate.

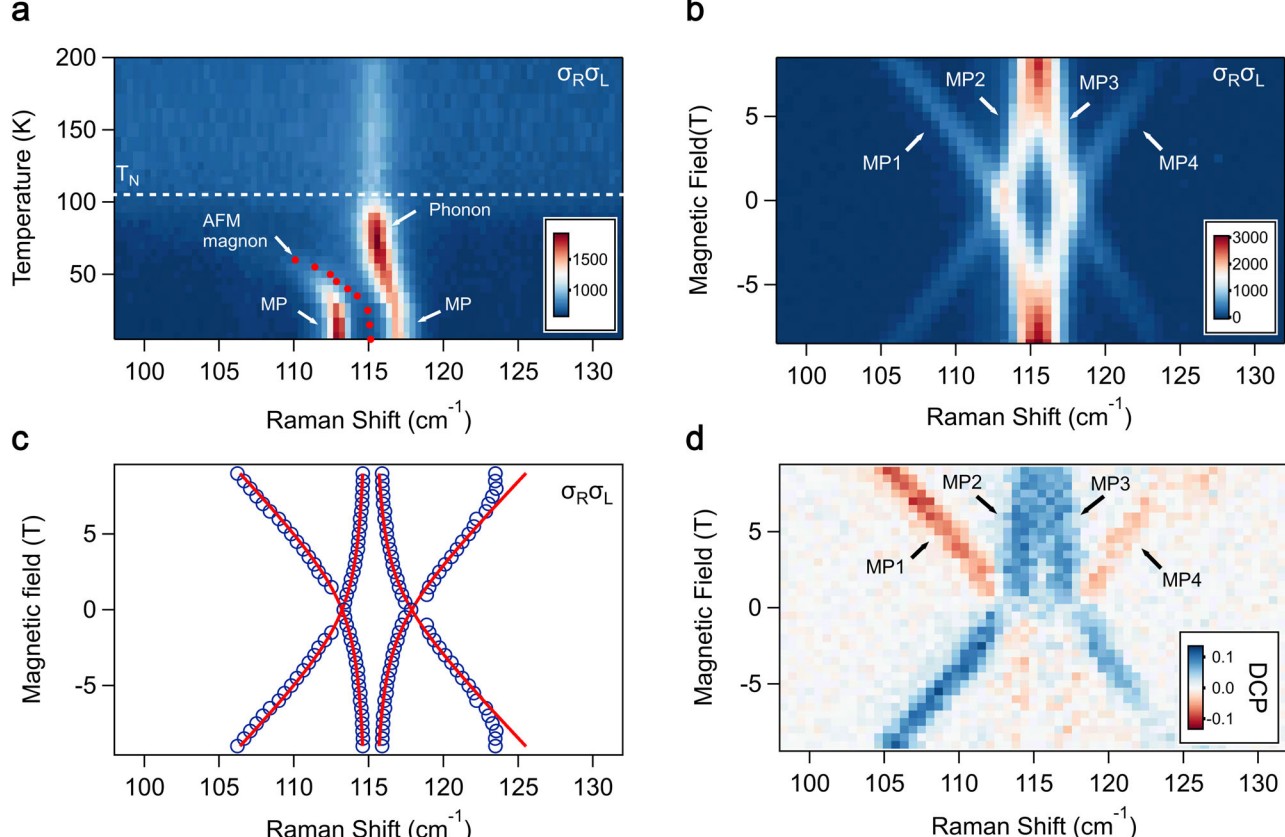

**Fig. 3 | Circular polarization-resolved magneto-Raman spectra of FePSe₃.**
**a** Temperature dependence of Raman spectra of a 96 nm FePSe₃ from 5 K to 200 K. The two prominent modes at low temperature are magnon polarons (MP). As the temperature increases the AFM magnons red shift, which enlarges the magnon-phonon detuning and leads to the redistribution of Raman intensity among the two MP modes. The red dots are extracted uncoupled AFM magnon frequencies.

**b** Magneto-Raman spectra (from 9 T to −9 T) with right-handed circular excitation and left-handed detection ($\sigma_R\sigma_L$). **c** Peak positions of the MP modes (blue circles) fitted with the selective strong coupling model (red lines). The deviation of the fit -125 cm⁻¹ is due to the avoided crossing with nearby phonons. **d** Degree of circular polarization (DCP) spectra of FePSe₃, with the DCP defined as ($\sigma_R\sigma_L - \sigma_L\sigma_R$)/ ($\sigma_R\sigma_L + \sigma_L\sigma_R$). All MP branches exhibit circular polarization.

mass and the displacement for atom $\alpha$ at unit cell $i$, and from the symmetry of the phonon modes.

The coupling between magnons and phonons can be derived by considering a spin Hamiltonian where the magnetic interaction parameters $\mathbf{J}_{ij}$ depend on the ionic displacements $\mathbf{u}_i$ (see Supplementary Text S6). Expanding $\mathbf{J}_{ij}$ to lowest order in $\mathbf{u}_i$ results in the Hamiltonian

$$H_{m-ph} = -\frac{1}{R^2}\sum_{\langle ij \rangle}(\mathbf{u}_{ij}\cdot\mathbf{R}_{ij})\mathbf{S}_i\cdot([\boldsymbol{\alpha}\mathbf{J}]_{ij}\mathbf{S}_j),\qquad(1)$$

where the matrix $\mathbf{J}_{ij}$ encodes the coupling of spins $\mathbf{S}_i$ and $\mathbf{S}_j$, $\mathbf{u}_{ij} = \mathbf{u}_j - \mathbf{u}_i$ is the ionic displacement from equilibrium, $\mathbf{R}_{ij} = \mathbf{R}_i - \mathbf{R}_j$ is the equilibrium lattice vectors, and $R = |\mathbf{R}_{ij}|$. The matrix $\boldsymbol{\alpha}_{ij}$ is proportional to the derivative of $\mathbf{J}_{ij}$ with respect to $\mathbf{u}_i$, and quantify the strength of the magnon-phonon interaction. The formation of MPs requires a linear magnon-phonon coupling, which in FePSe₃ arises from anisotropy terms like $J_{xz}S_i^x S_j^z$ and $J_{yz}S_i^y S_j^z$ since the spins point along the $z$-axis.

An analysis of the coupling Hamiltonian shows that the selective coupling of magnons and chiral phonons follows from the symmetries of the FePSe₃ crystal space group $C2/m$. On the one hand, the non-symmorphic screw axis symmetry of FePSe₃ requires terms of the form $u_{ix}S_i^x S_j^z$ and $u_{iy}S_i^y S_j^z$ to vanish. As a result, phonon modes polarized along the $x$-axis ($y$-axis) only couple to magnetic anisotropies of the form $S_i^y S_j^z$ ($S_i^x S_j^z$), implying that the modes $P_1$ and $P_2$ only couple to the magnons via modulation of $J_{xz}$ and $J_{yz}$, respectively. On the other hand we note that Raman-active modes are even under parity, which ensures that the magnon-phonon coupling strengths satisfy $g_{i\alpha} = g_{i\beta}^*$, where

$i = 1, 2$ is the phonon index and $\alpha/\beta$ are indexes for the magnons. Since couplings proportional to $J_{xz}$ ($J_{yz}$) give a real (imaginary) contribution to the magnon-phonon coupling strength $g$, these observations show that $g_{1\alpha} = g_{1\beta}^* = g$ and $g_{2\alpha} = g_{2\beta}^* = ig$. Consequently, the coupling strength between the chiral phonons $P_\pm$ and the magnon $M_\alpha$ is $g_{\pm\alpha} = (g_{1\alpha} \pm ig_{2\alpha})/\sqrt{2}$, which leads to $g_{+\alpha} = \sqrt{2}g$ and $g_{-\alpha} = 0$. Similarly, for $P_\pm$ and the magnon $M_\beta$, we have $g_{+\beta} = 0$ and $g_{-\beta} = \sqrt{2}g$. Hence, the modes $P_\pm$ selectively couple to the AFM magnon that carries angular momentum in the same direction, as illustrated in Fig. 5b. Consequently, the four MP modes are chiral.

The symmetry arguments are supported by our first principle calculations, and the calculated MP branches are in good agreement with the experimental results, as shown in Fig. 5c. The calculated magnitude of the magnon-phonon coupling strength for $P_1$ and $P_2$ is 2 cm⁻¹ and 2.16 cm⁻¹, respectively, as listed in Table 1. They are in excellent agreement with the experimental values.

We note that without magnon-phonon coupling, the two (nearly) degenerate phonon modes can equally well be considered as either a superposition of two linearly polarized phonons or two circularly polarized phonons. They would remain nearly unchanged under external fields due to the negligible orbital phonon magnetic moments. It is the magnon-phonon coupling that effectively lifts the degeneracy between the two circularly polarized phonons with opposite angular momentum under magnetic fields. This happens because of the selective form of the coupling, where the right-handed (left-handed) magnon couples to the right-handed (left-handed) phonon. The magnon-phonon coupling is thus what induces the phonon chirality.

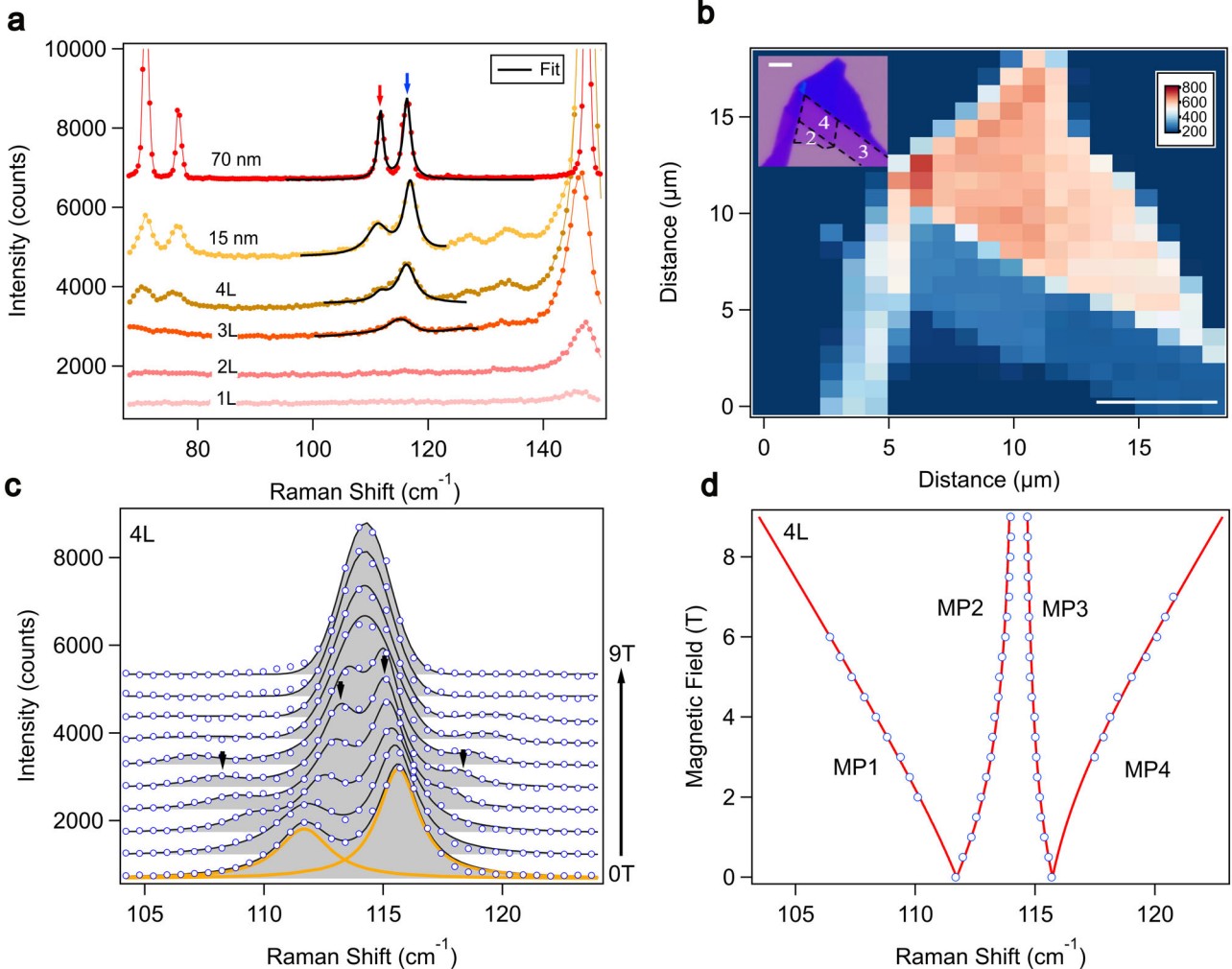

**Fig. 4 | Selectively coupled magnons and chiral phonons in the atomically thin FePSe$_3$ flakes revealed by Raman microscopy. a** Layer-dependent Raman spectra of FePSe$_3$ at 4 K, with the magnon polarons indicated by arrows. The black lines are fits to the MP modes. **b** Raman intensity mapping of the low-energy MP mode (indicated by the red arrow in panel **a**) of a FePSe$_3$ flake. Inset: the corresponding optical image of the flake. The scale bar equals 5 μm in length. **c** Magneto-Raman spectra of a 4L flake as a function of magnetic fields. All four MP branches are resolved as indicated by arrows. Black solid lines are the Lorentzian fitting curves. Orange lines are the fitting curves for individual MP modes. **d** Peak positions of MP branches (open circles) of the 4L flake, which can be well described by the selective strong coupling model (red lines).

To elucidate the unusual Raman circular polarization of the MP branches, we further calculated the Raman cross-sections of each scattering channel. The obtained DCP spectrum is shown in Fig. 5d, and matches well with the experimental data. We find that the DCP of each MP branch is determined by a delicate interplay of the Raman amplitudes of the bare magnons and phonons (see Supplementary Text S8). In particular, depending on the sign of the coupling constant $g$, either the lower or upper MP modes change polarization as a function of the magnetic field. This is due to a destructive interference between the contributions of the bare magnons and phonons to the Raman signal of the MPs, and explains the unusual DCP spectra observed in FePSe$_3$ (see Fig. 5d). We note that for large magnetic fields the DCP of the phonon modes gradually vanishes, while the polarizations dependence of the magnons revert to the behavior expected in the uncoupled limit, as shown in Fig. 5e.

In summary, we discover a unique selective strong coupling between magnons and chiral phonons in atomically thin FePSe$_3$. The energy splitting of the chiral phonons in a magnetic field mainly originates from their selective coupling with the AFM magnons, which exhibit clear Zeeman splitting. These findings establish 2D vdWs AFMs as an ideal playground for exploring novel chiral spin-lattice excitations,

in particular for honeycomb spin lattices. The selective hybridization of magnons and chiral phonons provides opportunities for coherent manipulation of spin and lattice angular momentum with circularly polarized light at an ultrafast timescale. We envision that the strong coupling between magnons, chiral phonons and circular photons may open a door toward entangled magnon-phonon-photon cavity systems for angular momentum encoded quantum information and simulation.

## Methods

### Crystal growth and sample fabrication

Single crystals of FePSe$_3$ were synthesized by the chemical vapor transport (CVT) method using iodine as the transport agent. Stoichiometric amounts of Iron powder (99.998%), phosphorous powder (98.9%), and sulfur pieces (99.9995%) were mixed with iodine (1 mg/cc) and sealed in quartz tubes (10 cm in length) under a high vacuum. The tubes were placed in a horizontal one-zone tube furnace with the charge near the center of the furnace. Sizeable crystals (10.0 × 10.0 × 0.5 mm³) were obtained after gradually heating the precursor to 750 °C, dwelling for a week, and cooling down to room temperature. Atomically thin FePSe$_3$ flakes were prepared on Si substrates with a layer of 285 nm SiO$_2$ via mechanical exfoliation. After

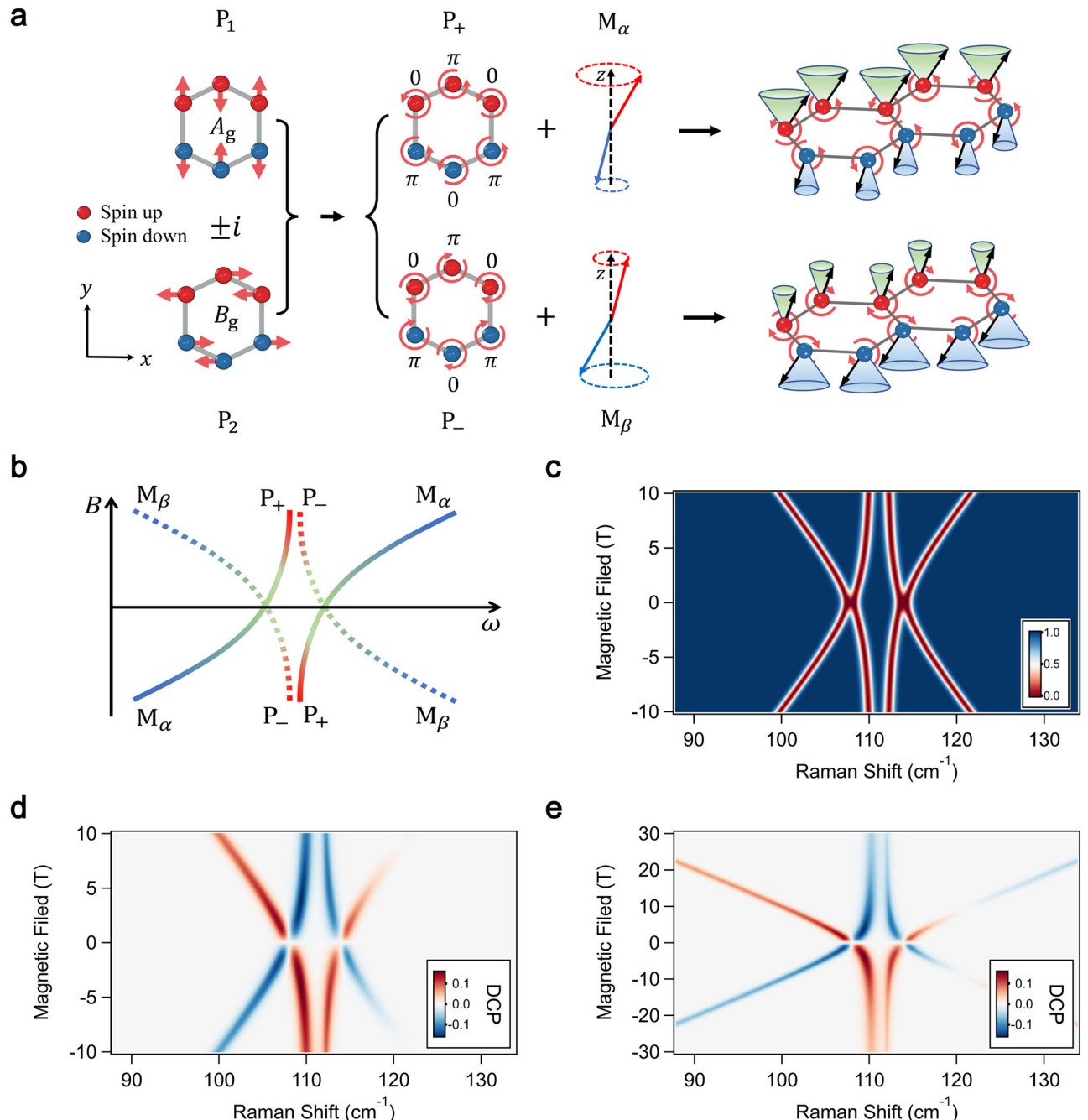

**Fig. 5 | First principle calculations of selective hybridization between magnons and chiral phonons in FePSe₃.** **a** Schematic diagrams for atomic displacement of P₁ and P₂ phonons, the resulting chiral phonons (P±), and their selective hybridization with AFM magnons (Mα and Mβ). Calculated atomic displacement of Fe atoms of P₁ and P₂ phonons are shown on the left. The displacements of P₁ and P₂ modes exhibit out-of-phase motions between both inter- and intra-zigzag chain nearest neighbors. Their superposition gives two chiral phonons P+ and P−, which selectively couple with the two AFM magnons. **b** Schematic diagram of the magnon

polaron branches under a magnetic field, resulting from two pairs of strongly coupled magnons and phonons given by (P+, Mα) and (P−, Mβ). **c**, Calculated magnon polarons dispersion as a function of magnetic field, which shows excellent agreement with optical results. **d** Calculated degree of circular polarization (DCP) of Raman spectra. The DCP of the magnon polarons is determined by a delicate interplay of the bare magnons and phonons, and the resulting interference of their Raman coefficients. **e** Calculated DCP spectra up to 30 T.

exfoliation, the samples were kept in a vacuum desiccator to prevent degradation. All the measurements were performed in a vacuum or Helium gas environment.

### Far-infrared magnetospectroscopy

The far-infrared transmission measurements were performed at the National High Magnetic Field Laboratory in Tallahassee, FL, using a combination of FT-IR spectrometer with 17 T superconducting and

35 T resistive magnets. The far-IR radiation was propagated inside an evacuated optical beam line from the spectrometers to the top of the magnets. Thereon, the light brass pipe guided the radiation down to the sample located at the center of the magnet and cooled down to 5 K by low-pressure helium exchange gas. The transmission was detected by a 4.2 K Si bolometer placed in the infinitesimal fringe field, just a short distance from the sample. The Faraday geometry was employed in all transmission measurements. To unveil field-dependent

excitations from those that are field-independent, spectra at each magnetic field step were divided by the average of all spectra, resulting in the successful suppression of field-independent background in the transmittance.

## Optical linear dichroism

The LD measurements were carried out in a reflection geometry in an optical cryostat. A 633 nm HeNe laser was double-modulated by a photoelastic modulator (PEM) and a mechanical chopper, and focused on the sample flakes by an objective. The beam size was <2 μm. The reflected light was detected by a photodiode and then demodulated by a lock-in amplifier. By rotating a halfwave waveplate, the reflection intensity pattern as a function of polarization angle was obtained.

## Polarization-resolved magneto-Raman spectroscopy

Raman measurements were carried out in a superconducting magnet at cryogenic temperatures down to 1.6 K. Magnetic fields (−9 T to 9 T) were applied in the Faraday geometry, which is parallel with the spin-pointing direction of FePSe$_3$. The excitation laser power was kept below 200 μW to avoid sample heating. A vacuum-compatible non-magnetic objective was utilized to focus the laser beam down to 1 μm. The back-scattered Raman signal was collected and sent to a high-resolution spectrometer with liquid-nitrogen-cooled CCD detector arrays. For polarization-resolved Raman detection, zero-order half- and quarter-waveplates were used.

## First principle calculations

To parameterize the Hamiltonian of Eq. S3 we performed first principle calculations with the Abinit electronic structure code[45–48]. We employed the DFT+$U$ formalism in the local density approximation using projector augmented wave (PAW) pseudopotentials, a plane wave cut-off of 20 Ha and 40 Ha for the plane wave and PAW parts, and included a Hubbard $U$ and Hund's $J$ of 3.5 eV and 0.25 eV calculated with the Octopus code via the ACBN0 functional[49,50]. A Γ-centered Monkhorst-Pack grid with dimensions $8 \times 6 \times 8$ was used to sample the Brillouin zone. The magnetic ground state was found to have a zigzag antiferromagnetic order with magnetic moments 3.22 $\mu_B$ almost completely aligned along the $z$-axis. The DFT + $U$ Bloch functions were mapped onto maximally localized Wannier orbitals via the Wannier 90 code[51,52], including the $d$-orbitals of Fe and the $p$-orbitals of P and Se in the description. Subsequently, the magnetic parameters were calculated by the Python package TB2J[53] implementing the KKR Green's function method for local spin rotation perturbations relying on the magnetic force theorem. The phonon frequencies and polarization vectors at the Γ point were calculated using density functional perturbation theory (DFPT) as implemented in Abinit, assuming a ferromagnetic interlayer coupling. The atomic positions and stresses were relaxed to $10^{-6}$ Ha/Bohr. The calculated phonon modes are in good agreement with previous DFT calculations and Raman scattering data. The magnon-phonon couplings were found by comparing the magnetic parameters calculated in the equilibrium state and in a state with the ionic coordinates displaced according to the phonon polarizations.

## Data availability

All data that support the plots within this paper and other findings of this study are available upon request.

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

## Acknowledgements

We thank Prof. Di Xiao for valuable discussion. We thank Qijun Zong and Lei Wang for their help of atomic force microscope measurements. This work is supported by the MOST of China (Grant No. 2020YFA0309200) and the Fundamental Research Funds for the Central Universities (0204-14380184), the Cluster of Excellence "CUI: Advanced Imaging of Matter" of the Deutsche Forschungsgemeinschaft (DFG), EXC 2056 (Project ID 390715994), and the Grupos Consolidados (Grant No. IT1249-19). The Flatiron Institute is a division of the Simons Foundation. A portion of this work was performed at the National High Magnetic Field Laboratory, which is supported by National Science Foundation Cooperative Agreement No. DMR-1644779 and the State of Florida. Bulk crystal growth is supported by grant no. NSF MRSEC DMR-1719797 and the Gordon and Betty Moore Foundation's EPiQS Initiative, Grant No. GBMF6759 to J.H.C. EVB acknowledges funding from the European Union's Horizon Europe research and innovation programme under the Marie Skłodowska-Curie grant agreement No. 101106809.

## Author contributions

J.C. and E.V.B. contributed equally to the work. Q.Z. designed the project. Q.J., J.H.C., C.L., and F.L. synthesized single crystal samples. J.C. prepared the thin-film samples via mechanical exfoliation and carried out the magneto-Raman measurements assisted by F.W. under the supervision of Q.Z.; M.O. performed the magneto-infrared spectroscopy in the Maglab. E.V.B. performed the first principle calculations, linear spin wave calculations as well as the analytical analysis of magnon-phonon coupling under the supervision of A.R.. J.C., Q.Z., M.O., and E.V.B. analyzed all the data and discussed with X.X. and A.R.; Q.Z., E.V.B., and J.C. co-wrote the manuscript with input from all other authors.

## Competing interests

The authors declare no competing interests.
