## [Peer Review File · Nature Communications]

Reviewers' Comments:

Reviewer #1:

Remarks to the Author:

I have read the manuscript by Cui et al. with great interest. The manuscript reports infrared optical and Raman measurements under external magnetic field, to identify the chirality of phonon/magnon excitations. The manuscript further argues selective magnon-phonon hybridization and magnon-induced chiral phonons in antiferromagnet (AFM) FePSe₃. Overall, the manuscript conveys important concept rather clearly, with decent agreement with the theoretical expectations and experimental results. I would like to recommend publication of the manuscript with some suggestions listed below.

1. I am not sure that "atomically thin" in the title and main text suitably represents the importance of the manuscript. The most important results is on bulk flake. It is shown that the thinner flakes do show some reminiscence of the feature, but they are definitely weaker. I do understand that the keyword "atomically thin" is attractive, but the manuscript would become more rigorous and focused without the keyword. I strongly suggest removing the keyword and most of the discussion regarding "atomically thin" in the title, abstract and introduction.

2. I would appreciate if the authors could further explain the minute discrepancy between the theoretical calculation and the experimental data (Figs. 3d and 5d). For example, the experimental Raman spectra with a positive magnetic field show that the negative DCP of MP1 is much larger than MP4 (Fig. 3d), calculated MP dispersion shows the opposite DCP trend (Fig. 5d). Similar opposite behavior is also shown for the negative magnetic field region.

3. The phonon Zeeman effect has been identified in previous publications both experimentally and theoretically, suggestive of angular momentum of phonons, e.g., *Sci. Adv.* 8, eabm4005 (2022), *Phys. Rev. Lett.* 119, 075301 (2017), *Phys. Rev. Materials* 1, 014401 (2017). Therefore, current manuscript would read much clearer, if it could explain the necessity of introducing magnon-phonon coupling, instead of simply discussing chiral phonons. Additionally, one might further ask, 1) why linear magnon-phonon coupling is required for MP Formation and 2) justification of dot product between ionic displacement and lattice vector in the Hamiltonian.

4. Out of curiosity, although the authors focused on an antiferromagnetic system, can magnon-phonon coupling be realized in a ferromagnetic system? What is the requirement for the magnon-phonon hybridization? What would be the role of spin-orbit coupling in this system? These discussions might help further generalize the implications of the manuscripts.

Reviewer #2:

Remarks to the Author:

Chirality selective magnon-phonon hybridization and magnon-induced chiral phonons in an atomically thin antiferromagnet

The authors study the interplay between magnons and chiral phonons, and the details of chiral phonon formation in a magnetic system. The main claim is the observation of magnon-induced chiral phonons and chirality selective magnon-phonon hybridization in an atomically thin 2D antiferromagnet FePSe₃. The research topic is interesting and extremely difficult. Demonstration of chiral phonons by itself is a major task and typically not all experts are convinced. Adding magnon phonon hybridization on top of chiral phonons makes the task formidable. The experiments and theory are performed at a high technical level and the paper is well-written. I have the following technical questions for the authors to clarify.

1) What is the criterion of the magnon-phonon coupled state? At what point we can say that these are not strongly interacting phonons and magnons but a new coupled state? What are the general properties (statistics, dispersion, localization, binding energy, etc.) of the hybrid quasiparticles magnon polarons?

2) Why atomic (2D) limit is important for the demonstration of the chirality selective magnon-phonon hybridization? Why it cannot be done with thicker films?

3) How the local excitation laser heating affects the results? What is the laser power on the surface of the flakes? The claimed accuracy of the Raman measurements here is a fraction of the cm^{-1} (see table I). The local laser heating can produce larger shifts for typical Raman experiment power levels.

Reviewer #3:

Remarks to the Author:

In the manuscript by Cui and co-workers, the authors demonstrate the emergence of magnon-polarons in a two-dimensional antiferromagnet. Investigating the excitations using different optical techniques, they determine the relevant eigenmode energies as a function of applied magnetic field and temperature. The spectra thus recorded manifest a clear level repulsion-induced gap. Furthermore, the dependence of the spectra on the applied magnetic field fits well with a theoretical model that considers chirality selective hybridization between the two degenerate magnon modes and two degenerate phonon modes. Further insights into the nature of the magnon-phonon coupling have been provided using first-principle calculations.

This is a high quality work addressing the urgent topic of engineering magnon-polarons in two-dimensional magnets. The methods employed, both experimental and theoretical, are adequate and provide solid evidence in favor of the authors' claims. Therefore, I recommend publication of this manuscript in Nature Communications after the authors address the following minor concerns.

1. In Fig. 1b, the linear dichroism signal does not vanish at the Neel temperature. It survives above it. More importantly, close to the assumed Neel temperature, the linear dichroism signal deviates significantly from the expected temperature dependence of the Neel order squared. Can the authors comment on this deviation?

2. Figure 2a forms the theoretical basis for a phenomenological understanding of the experimental observations. Here, the authors assume that the two magnon modes are degenerate in the absence of their coupling with the phonons. In other words, one of the main assumptions that allows attributing the experimentally observed hybridization gap to magnon-polaron formation is that there is no direct coupling between the two magnon modes. However, any anisotropy that breaks the axial symmetry about the z axis will couple the magnon modes as detailed in Phys. Rev. B 96, 020411(R) (2017). Such an anisotropy can be expected especially in this honeycomb lattice. Hence, I would suggest the authors to consider and rule out a potential role of the direct magnon-magnon coupling in their observed phenomena.

3. In the caption to Fig. 4, the authors state "The selective hybridization of magnons and phonons is clearly observed in the 4L samples." This is not so clear to me. I see clear theory curves that do not correspond to the experimental data so well. Could the authors clarify?

4. The authors explain the chirality selective hybridization between the magnon and phonon modes via various mathematical arguments. Naively, we could say that the linearized magnetoelastic coupling (which is responsible for magnon-polaron formation) preserves the isotropy and hence the angular momentum conservation. Thus, the selectivity of the coupling is simply a statement of angular momentum conservation between the hybridizing magnon and phonon modes. Could the authors agree or disagree with this argument stating their reasoning?

Response to reviews for "Chirality selective magnon-phonon
hybridization and magnon-induced chiral phonons in a layered zigzag
antiferromagnet"

We are grateful for the referees' careful review of our manuscript and for their comments, questions, and suggestions, which we address below and in the revised manuscript. Referee remarks are in *blue italic font*.

Report of referee #1

I have read the manuscript by Cui et al. with great interest. The manuscript reports infrared optical and Raman measurements under external magnetic field, to identify the chirality of phonon/magnon excitations. The manuscript further argues selective magnon-phonon hybridization and magnon-induced chiral phonons in antiferromagnet (AFM) FePSe₃. Overall, the manuscript conveys important concept rather clearly, with decent agreement with the theoretical expectations and experimental results. I would like to recommend publication of the manuscript with some suggestions listed below.

Our reply #1:

We appreciate the referee for carefully reading our manuscript and providing valuable comments and suggestions. We followed the advice and improved our paper. In the following response, we address these comments and questions in a point-by-point manner.

I am not sure that "atomically thin" in the title and main text suitably represents the importance of the manuscript. The most important results is on bulk flake. It is shown that the thinner flakes do show some reminiscence of the feature, but they are definitely weaker. I do understand that the keyword "atomically thin" is attractive, but the manuscript would become more rigorous and focused without the keyword. I strongly suggest removing the keyword and most of the discussion regarding "atomically thin" in the title, abstract and introduction.

Our reply #2:

We thank the referee for the suggestion. Indeed, the chirality selective magnon-phonon coupling is observed in both bulk flakes and atomically thin samples. We agree that the essential physics remains the same in both cases. Therefore, we follow the referee's suggestion and removed the term "atomically thin" in the title, abstract, and introduction. The new title is "Chirality selective magnon-phonon hybridization and magnon-induced chiral phonons in a layered zigzag antiferromagnet". We note that the magnon mode and the phonon motions involved in the selective strong coupling are two-dimensional in nature, and the symmetry of the zigzag antiferromagnetic order plays a central role in protecting the selective coupling between phonon and magnon with the equal direction of

angular momentum.

Changes made:

- Change title to “Chirality selective magnon-phonon hybridization and magnon-induced chiral phonons in a layered zigzag antiferromagnet”
- Removed ‘atomically thin’ in the abstract and introduction.

I would appreciate if the authors could further explain the minute discrepancy between the theoretical calculation and the experimental data (Figs. 3d and 5d). For example, the experimental Raman spectra with a positive magnetic field show that the negative DCP of MP1 is much larger than MP4 (Fig. 3d), calculated MP dispersion shows the opposite DCP trend (Fig. 5d). Similar opposite behavior is also shown for the negative magnetic field region.

Our reply #3:

We are very thankful to the referee for noticing this point since it allowed us to identify a mistake in our calculation. The behavior noted by the referee is determined by the sign of the electron-phonon coupling g , as discussed in Supplementary Section 8. Depending on the sign of g , either the upper or lower magnon branch shows a cross-over in its polarization dependence as a function of the magnetic field B . We had made a mistake in the calculation of the degree of circular polarization (DCP) where the sign of g had been switched. In the revised manuscript these results have been corrected, and now agree much better with the experimental data.

Fig. R1 The corrected Raman circular polarization calculations of the magnon polarons in FePSe₃. The calculated degree of circular polarization (DCP) of the MP1 and MP4 branches qualitatively agrees with the experimental results.

The other reason that caused the discrepancy between the measured and calculated DCP of the MP4 is an additional magnon-phonon hybridization around 125 cm⁻¹. As shown in the dotted area in both Fig. R2a and c, an optical phonon mode at 125 cm⁻¹ hybridizes with the MP4 branch revealed by both FIR and Raman spectra. Such hybridization modifies the Raman circular polarization selection of MP4, while MP1 is less affected. The phonon mode at 125 cm⁻¹ will borrow some oscillator strength from MP4, thus reducing the signal on this side of the spectrum, an effect that is not included in our calculations, where we mainly focus on the magnon polaron branches hybridized from phonons around 115 cm⁻¹ and the AFM magnons for simplicity. That is likely the other reason

we see the small discrepancy between the measured and the calculated DCP of MP4.

Fig. R2. The hybridization of MP4 and the optical phonon at 125 cm^{-1} revealed by Far-infrared transmission and magneto-Raman spectra. a, FIR spectra. The dotted area indicates hybridization between the optical phonon mode at 125 cm^{-1} and the MP4 branch. **b**, Peak positions of MP branches (blue circles) fitted with selective strong coupling model (orange lines). The deviation of the MP4 fitting line from the experimental peak positions is due to the hybridization with the phonon at 125 cm^{-1} . **c**, Magneto-Raman spectra. **d**, Peak positions of the MP modes in Raman.

Changes made:

- Modified Fig. 5d and 5e with corrected Raman calculation results in the main text.

The phonon Zeeman effect has been identified in previous publications both experimentally and theoretically, suggestive of angular momentum of phonons, e.g., Sci. Adv. 8, eabm4005 (2022), Phys. Rev. Lett. 119, 075301 (2017), Phys. Rev. Materials 1, 014401 (2017). Therefore, current manuscript would read much clearer, if it could explain the necessity of introducing magnon-phonon coupling, instead of simply discussing chiral phonons. Additionally, one might further ask, 1) why linear magnon-phonon coupling is required for MP Formation and 2) justification of dot product between ionic displacement and lattice vector in the Hamiltonian.

Our reply #4:

We thank the referee for the question. We agree that the phonon Zeeman splitting can be utilized to identify the chiral phonons if they possess relatively large magnetic moments. However, for regular phonons, even if they carry finite angular momentum, the orbital magnetic moments of phonons are

negligibly small, around 10^{-3} to 10^{-4} Bohr magneton (μ_B), which is due to the large ion mass with respect to electrons [Phys. Rev. Mater. 3, 064405 (2019)]. Sizeable phonon Zeeman splitting in the previous discussions is due to either the coupling between phonon and 4f crystal-field excitations [J. Phys. C: Solid State Phys. 9, L297 (1976), Phys. Rev. Research 4 013129 (2022)], or the coupling between phonons and topologically nontrivial electronics bands [Phys. Rev. Lett. 119, 075301 (2017), Phys. Rev. Lett. 127, 186403 (2021), Phys. Rev. Lett. 128 075901 (2022)]. None of these couplings exists in FePSe₃.

In the case of FePSe₃, without magnon-phonon coupling, the two relevant phonon modes can equally well be considered as either a superposition of two linearly polarized phonons or two circularly polarized (chiral) phonons. This arbitrariness is due to the degeneracy of the phonon modes, arising from the E_g representation of the D_{3d} point group. Without magnon-phonon coupling, the phonon modes would remain nearly unchanged under external fields, and no phonon Zeeman splitting would be resolved, due to the negligibly small orbital phonon magnetic moments.

Due to the magnon-phonon coupling, the degeneracy between the left- and right-handed circularly polarized phonons is effectively lifted. This happens because of the selective form of the coupling, where the right-handed (left-handed) magnon couples to the right-handed (left-handed) phonon. Effectively this decouples the right- and left-handed sectors of the theory, making an interpretation of the phonons as chiral necessary. The magnon-phonon coupling is thus what generates the phonon chirality. We note that without the coupling, the magnons and phonons in FePSe₃ are nearly resonant at zero magnetic field. Therefore, the external field not only changes the Zeeman energy of magnons and phonons, but also strongly affects their detuning. Hence, instead of a linear phonon Zeeman splitting, in this case, an avoided crossing behavior is expected, as also observed in the experiment.

In summary, it is the magnon-phonon coupling that generates the phonon chirality, and which is responsible for the phonon (or magnon polaron) modes splitting under magnetic fields. The magnon-phonon coupling is an essential point in understanding the phonon chirality and all experimental observations in the present context.

why linear magnon-phonon coupling is required for MP Formation

The Hamiltonian of the coupled magnon phonon system should have the following form,

$$H = \hbar\omega_{mag}a^\dagger a + \hbar\omega_{ph}b^\dagger b + H_{coupling},$$

where a (b) is the annihilation operator of magnons (phonons). Magnon polarons (MP) are linearly hybridized magnons and phonons, and are the new eigenmode of the coupled system.

$$H = \hbar\omega_{mag}a^\dagger a + \hbar\omega_{ph}b^\dagger b + H_{coupling} = \hbar\omega_{MP}c^\dagger c$$

where c (c^\dagger), the annihilation (creation) operator of magnon polaron, is a linear combination of magnon and phonons operators. Therefore, $H_{coupling}$ should have the quadratic form like, $(a^\dagger + a)(b^\dagger + b)$, which is linear for individual magnon and phonon operators. In contrast, terms like $a^\dagger a(a^\dagger + b)$ renormalizes the magnon frequency rather than provides hybridization and avoided crossing. Therefore, a linear magnon-phonon coupling is necessary for MP formation.

justification of dot product between ionic displacement and lattice vector in the Hamiltonian.

The dot product between the ionic displacement and the lattice vector arises from a straightforward

Taylor expansion of the magnetic exchange J in the relative displacement $u_{ij} = \mathbf{R}_{ij} - \mathbf{R}_{ij}^0$, together with the assumption that the magnetic exchange interaction only depends on the distance $u_{ij} = |u_{ij}|$.

Changes made:

- Added a short paragraph in the main text highlighting the central role of magnon-phonon coupling in the formation of chiral phonons.

Out of curiosity, although the authors focused on an antiferromagnetic system, can magnon-phonon coupling be realized in a ferromagnetic system? What is the requirement for the magnon-phonon hybridization? What would be the role of spin-orbit coupling in this system? These discussions might help further generalize the implications of the manuscripts.

Our reply #5:

We thank the referee for these insightful questions. Yes, the magnon-phonon strong coupling is allowed in ferromagnetic (FM) systems and has been studied both theoretically and experimentally. Historically, the coherent coupling between FM spin waves and traveling acoustic phonons was first considered by Prof. Kittel [Phys. Rev. 110, 836 (1958)], where the coupling results from the phonon modulation of exchange interactions at finite wavevector k . Experimentally, the FM magnons match with acoustic phonons in energy. Several experimental approaches have been utilized to identify the hybridization, for instance, transport measurement based on the spin Seebeck effect [Phys. Rev. Lett.117.207203 (2016)], ultrafast pump-probe technique [Phys. Rev. B.102.144438 (2020)], and cavities and surface patterning [Sci. Adv. 1501286 (2016), Phys. Rev. B.102.144438 (2020), Nat. Communi. 10, 2652 (2019), Phys. Rev. B. 96,100406 (2017)]. Recently theoretical efforts draw additional attention to the topological aspect of the magnon-phonon coupling, since such coupling may enable finite Berry curvatures around the hybridization area of the magnonic bands, and further lead to topologically nontrivial magnonic effects [Phys. Rev. Lett.117.217205 (2016), Phys. Rev. Lett.123.237207 (2019), Phys. Rev. B.99.174435 (2019)].

‘What is the requirement for the magnon-phonon hybridization?’

Phenomenologically, linear magnon-phonon interaction is required. In contrast, a quadratic coupling in either the magnons or the phonons would instead give rise to magnon-phonon scattering, as we discussed in Reply #4. From the symmetry viewpoint, the involved magnon and phonon modes need to belong to the same irreducible representation in order to hybridize. From the practical viewpoint, the energy of magnon and phonon modes needs to be close, (or can be tuned to be close). The magnon phonon detuning should be comparable with the coupling strength to have clear avoided-crossing behavior between magnon and phonon bands.

‘What would be the role of spin-orbit coupling in this system?’

For the FePSe₃ system, spin-orbit coupling plays an essential role in magnon-phonon hybridization. As discussed above, linear magnon-phonon coupling is necessary for the formation of magnon polarons. As a collinear AFM system, the spins in FePSe₃ are along the z-direction. Thus, to give the $(a^\dagger + a)(b^\dagger + b)$ type of coupling terms, it requires the coupling arising from magnetic anisotropies proportional to $S^x S^z$ or $S^y S^z$. These anisotropic terms are typically generated by spin-

orbit coupling (although they can also arise e.g., from magnetic dipole-dipole interactions), and can be either symmetric (as in the present case) or antisymmetric (as for a Dzyaloshinskii-Moriya interaction). As discussed in Supplementary Section S2, two-dimensional magnets with an out-of-plane magnetic order cannot have magnetic anisotropies of the required form generated by dipole-dipole interactions, and thus for van der Waals magnets like FePSe₃ spin-orbit coupling is crucial. If we intentionally turn off the SOI in the first-principle calculation, the magnon-phonon hybridization vanishes in FePSe₃.

Report of referee #2

The authors study the interplay between magnons and chiral phonons, and the details of chiral phonon formation in a magnetic system. The main claim is the observation of magnon-induced chiral phonons and chirality selective magnon-phonon hybridization in an atomically thin 2D antiferromagnet FePSe₃. The research topic is interesting and extremely difficult. Demonstration of chiral phonons by itself is a major task and typically not all experts are convinced. Adding magnon phonon hybridization on top of chiral phonons makes the task formidable. The experiments and theory are performed at a high technical level and the paper is well-written. I have the following technical questions for the authors to clarify.

Our reply #6:

We appreciate the referee for the careful reading of the manuscript and the recognition of our discovery. We address the questions and comments in the following.

What is the criterion of the magnon–phonon coupled state? At what point we can say that these are not strongly interacting phonons and magnons but a new coupled state? What are the general properties (statistics, dispersion, localization, binding energy, etc.) of the hybrid quasiparticles magnon polarons?

Our reply #7:

The criterion of the formation of magnon polarons (MP) can be defined as the regime where the coupling strength g is larger than the decay rates for individual magnons and phonons (γ_{mag} and γ_{ph}). In this regime, the magnon and phonon bands are strongly coupled in a non-perturbative manner rather than slight renormalization of magnon/phonon energy (weak coupling). This criterion is similar to those for defining the formation of polaritons, the hybridized quasiparticles of photonic modes, and matter excitations in the strong coupling regime of light-matter interaction.

Quantitatively, one can define cooperativity $C = g^2/\gamma_{ph}\gamma_{mag}$. When the cooperativity $C > 1$, the couple is in the strong coupling regime, and the magnon polaron forms. When the cooperativity $C \leq 1$, the system is in the weak coupling regime. The weak interaction of magnons and phonons and their short lifetimes jeopardize the coherent hybridization of magnons and phonons, thus, they

cannot be viewed as new quasiparticles. In the experiment, such criterion translates into whether the energy gap between the upper and lower MP modes can be well resolved

What are the general properties (statistics, dispersion, localization, binding energy, etc.) of the hybrid quasiparticles magnon polarons?

As the hybridized state of phonons and magnons, the general properties of magnon polarons are somewhat the combination of these two types of excitations and strongly depend on the detuning (Δ). Regarding the statistics, since both magnons and phonons are bosons, thus, magnon polarons obey bosonic statistics. Regarding the dispersion, as schematically shown in Fig. R3, the magnon phonon hybridization changes with the wavevector, since these two types of quasiparticles typically exhibit different dispersion. Hence, the avoided crossing usually happens in a narrow range of the momentum space. The magnon polaron can be either localized or nonlocalized depending on the dispersion of the magnon and phonon bands. For instance, a recent study reports the observation of nonlocal magnon-polarons [Phys. Rev. B.96.104441(2017)].

In general, for large detuning, the properties of MPs recover those of pure magnons and phonons. But it is hard to define a particular point at which the MP states are again pure magnons and phonons, since different properties are not equally sensitive to the mode mixing. In particular, we find that the degree of circular polarization of Raman spectra shows signs of magnon-phonon mixing at very large detuning ($\Delta > 10g$), while the dispersion of the modes is only affected for $\Delta \sim g$, where g is the coupling strength.

Regarding the binding energy, MPs are the linear hybridization of magnons and phonons rather than their bound states, which is different from the cases of excitons or two-magnon bound states. However, one can still consider the coupling strength g as the ‘binding energy’ for magnon polarons, since it is exactly the energy difference between the lower MP and the bare magnon/phonon at zero detuning. The coupling strength g is on the order of 0.1 meV in FePSe₃.

Fig. R3. The schematic dispersion curves of magnon polarons. The curves with solid lines represent the dispersion of magnon polaron bands, while the red and blue dashed curves represent the uncoupled magnons and phonons, respectively.

Why atomic (2D) limit is important for the demonstration of the chirality selective magnon-phonon hybridization? Why it cannot be done with thicker films?

Our reply #8:

First, we want to clarify that the magnon-phonon hybridization can happen in both atomic limit and thicker films, as we demonstrated in 4L and bulk FePSe₃ flakes. To avoid confusion, we revised the title of the paper and remove the term “atomically thin”. The abstract and introduction are changed accordingly.

Second, we note that it is indeed important to demonstrate magnon-phonon hybridization and unusual chiral selectivity in the atomic (2D) limit. On one hand, recent theoretical studies suggested such hybridization in 2D magnetic lattice enables Berry curvatures in the avoid-crossing regions of the magnon and phonon bands, which lead to nontrivial magnon topology [Phys. Rev. Lett.117.217205 (2016), Phys. Rev. Lett.123.237207 (2019), Phys. Rev. B.99.174435 (2019), Phys. Rev. Lett.124.147204 (2020)], those results are sensitive to the 2D nature of the system. On the other hand, two-dimensional lattices, such as honeycomb and kagome, can host chiral phonons with finite angular momentum, which offers a fertile ground for exploring the interplay between chiral collective excitations. Our discovery provides one of the first examples in this direction. Moreover, the realization of MP states in atomically thin magnets readily opens up intriguing opportunities for manipulating MP with electrostatic gating, strain, interfacial engineering, and proximity effects. Those efforts will further benefit the design of novel spin-phononic devices based on coupled spin and lattice excitations. Therefore, we believe it is of great interest from both fundamental science and device applications viewpoints to study the magnon polaron in 2D systems.

How the local excitation laser heating affects the results? What is the laser power on the surface of the flakes? The claimed accuracy of the Raman measurements here is a fraction of the cm^{-1} (see table I). The local laser heating can produce larger shifts for typical Raman experiment power levels.

Our reply #9:

We agree with the referee that strong laser excitation may affect the formation of magnon polaron. In our experiment, the laser power was kept below 200 μ W to avoid sample heating (Fig.3a), which is weaker than the intensity used in regular Raman tests.

As we can see in Fig. R4, the local temperature increase will cause the redshift of magnon polaron, and the Raman intensity redistribution between the two MP peaks. We intentionally increase excitation power to 600 μ W for the same sample, the Raman spectrum is shown in Fig.R4. Compared with the 200 μ W data, there is no noticeable peak shift or broadening for the two MP modes, indicating negligible laser heating effects. Furthermore, we performed a rough estimation of sample temperature via the ratio of the anti-Stokes and Stokes intensities assuming a Boltzmann

distribution. For the 200 μW excitation, it gives a local temperature increase of 3 K. Moreover, by monitoring those zone-folded phonons in FePSe₃ Raman spectra, we can determine the Néel temperature of the system, which is consistent with the linear dichroism measurements and other previous reports, which further confirms the laser heating is marginal in our Raman measurements. Therefore, we conclude the laser heating effect for 200 μW excitation is negligibly small for the discussion of magnon polarons in our study.

Fig. R4. The Raman spectra of FePSe₃ with different excitation power. **a**, Raman spectra of Magnon polarons as a function of temperature at zero magnetic field. **b**, Raman spectra with 200 μW and 600 μW excitation laser power. No noticeable peak shift or linewidth broadening for the 600 μW spectrum with respect to the 200 μW one.

Report of referee #3

In the manuscript by Cui and co-workers, the authors demonstrate the emergence of magnon-polarons in a two-dimensional antiferromagnet. Investigating the excitations using different optical techniques, they determine the relevant eigenmode energies as a function of the applied magnetic field and temperature. The spectra thus recorded manifest a clear level repulsion-induced gap. Furthermore, the dependence of the spectra on the applied magnetic field fits well with a theoretical model that considers chirality selective hybridization between the two degenerate magnon modes and two degenerate phonon modes. Further insights into the nature of the magnon-phonon coupling have been provided using first-principle calculations.

This is a high quality work addressing the urgent topic of engineering magnon-polarons in two-dimensional magnets. The methods employed, both experimental and theoretical, are adequate and provide solid evidence in favor of the authors' claims. Therefore, I recommend publication of this manuscript in Nature Communications after the authors address the following minor concerns.

Our reply #10:

We are grateful for the positive evaluation of our research and we thank the referee for the valuable suggestions. We have addressed the questions and comments below.

In Fig. 1b, the linear dichroism signal does not vanish at the Neel temperature. It survives above it. More importantly, close to the assumed Neel temperature, the linear dichroism signal deviates significantly from the expected temperature dependence of the Neel order squared. Can the authors comment on this deviation?

Our reply #11:

We thank the referee for this insightful question. Indeed, the linear dichroism (LD) curve of FePSe₃ does not vanish right above the Néel temperature, instead, it exhibits a ‘tail-like’ behavior. We believe it may result from the following reasons:

- (1) When the long-range zigzag order disappears, short-range antiferromagnetic interactions still survive above Néel temperature. These incoherent short-range fluctuations may contribute to the tail signal of LD deviating from the quadratic linear dichroism dependence. This effect is expected to be more pronounced for two-dimensional systems compared to 3D cases, due to stronger spin fluctuation.
- (2) Another reason is that the AFM-to-paramagnetic transition in FePSe₃ may not be an ideal second-order phase transition. It is believed a lattice distortion is associated with the zigzag spin order, which makes the transition first order, and will certainly affect the critical behaviors of the order parameter near the critical temperature. Although the detailed X-ray scattering analysis is lacking for FePSe₃ across the T_N, such lattice distortion is systematically studied in a very similar compound, FePS₃ [Nat. Communi. 13, 1 (2022)], which also exhibits the tail-like LD curve above T_N [Nano Letters 21, 6938 (2021)]. We speculate the lattice distortion fluctuation, in an analogy of the nematic order fluctuation in iron pnictides, may contribute to the tail-like feature. This possibility is still under investigation.

Figure 2a forms the theoretical basis for a phenomenological understanding of the experimental observations. Here, the authors assume that the two magnon modes are degenerate in the absence of their coupling with the phonons. In other words, one of the main assumptions that allows attributing the experimentally observed hybridization gap to magnon-polaron formation is that there is no direct coupling between the two magnon modes. However, any anisotropy that breaks the axial symmetry about the z axis will couple the magnon modes as detailed in Phys. Rev. B 96, 020411(R) (2017). Such an anisotropy can be expected especially in this honeycomb lattice. Hence, I would suggest the authors to consider and rule out a potential role of the direct magnon-magnon coupling in their observed phenomena.

Our reply #12:

We thank the referee for raising this point. In fact, both the in-plane magnetic anisotropies and dipole-dipole interactions are very small in FePSe₃, on the order of 0.01 meV, as found from our first-principles calculations. The hybridization gap observed is around 0.25 meV, which is more than one order of magnitude larger than the energy scales responsible for magnon-magnon interactions. Thus, it is very unlikely that the observed magnon polaron peaks are due to magnon-magnon interactions. Furthermore, a magnon-magnon interaction would not explain the lifted degeneracy of the phonon modes, as well as the polarization dependence of the MP modes. Therefore, the simplest explanation of all the observed features of our observed results is a chirality selective magnon-

phonon coupling.

In the caption to Fig. 4, the authors state “The selective hybridization of magnons and phonons is clearly observed in the 4L samples.” This is not so clear to me. I see clear theory curves that do not correspond to the experimental data so well. Could the authors clarify?

Our reply #13:

We thank the referee for pointing this out. We improved our measurements on the 4L sample. The raw Raman spectra of the 4L FePSe₃ flake are shown in Fig.R5a. The phonon-like magnon polaron bands, MP2 and MP3, are well resolved as indicated by black arrows in Fig. R5a. One can also identify the two magnon-like MP modes, MP1 and MP4. The extracted peak positions of MP bands are demonstrated in Fig.R5b, which can be well described with the selective strong coupling model (red lines). We replaced the data in Fig.4 accordingly.

Fig. R5. Magneto-Raman spectra of the 4L FePSe₃ flake. a, Raw Raman spectra (open circles), and their Lorentzian fitting curves (black solid lines) as a function of magnetic fields. All four magnon polaron (MP) bands can be resolved as indicated by the arrows. Orange lines represent Lorentzian fitting for individual MP modes. **b,** Peak positions of MP branches (open circles) fitted with selective strong coupling model (red lines).

Changes made:

Update Figure 4 in the main text with new Raman spectra of the 4L FePSe₃, removing the colormap for the 3L sample.

The authors explain the chirality selective hybridization between the magnon and phonon modes via various mathematical arguments. Naively, we could say that the linearized magnetoelastic coupling (which is responsible for magnon-polaron formation) preserves the isotropy and hence the angular momentum conservation. Thus, the selectivity of the coupling is simply a statement of angular momentum conservation between the hybridizing magnon and phonon modes. Could the

authors agree or disagree with this argument stating their reasoning?

Our reply #14:

We thank the referee for the question. In qualitative terms, this argument is correct. However, it is not possible here to invoke strict angular momentum conservation. The main reason for this is that chiral phonons generally do not carry quantized angular momenta in terms of \hbar (for a further discussion see L. Zhang and Q. Niu, Phys. Rev. Lett. **112**, 085503 [2014] and Phys. Rev. Lett. **115**, 115502 [2015]), which also explains how linearly polarized phonons can hybridize with magnons (see e.g. S. Liu et al., Phys. Rev. Lett. **127**, 097401 [2021]). However, our results indicate that there is still a strong preference for magnons of a given angular momentum to hybridize with the corresponding phonons, which can be interpreted as a form of angular momentum selectivity (in contrast to conservation). And in our result, we rule out the most studied linear magnetoelastic coupling. The linear magnetoelastic coupling results from the phonon modulation of exchange interactions at finite wavevector k . In our experiment, due to the negligibly small photon momenta, the observed magnon-phonon strong coupling locates at the Γ point ($k = 0$), where the linear magnetoelastic coupling vanishes.

List of changes:

1. Line 1, change title to “Chirality selective magnon-phonon hybridization and magnon-induced chiral phonons in a layered zigzag antiferromagnet”
2. Line 28, change “an atomically thin 2D” to “a layered zigzag”
3. Line 38, remove “in the atomic limit and”
4. Line 70, remove “atomically thin”
5. Line 52, added Ref 27
6. Line 85, remove “in 2D”
7. Line 191-196, change “Figure 4c and 4d present the magnetic field dependence of the MPs in 4L and 3L flakes, respectively. The field-induced shift of MP2 and MP3 can be seen clearly. The red lines indicate fitted MP branches. The magnon-phonon coupling strength remains strong in 4L flakes reaching 2.0 cm^{-1} , which is very close to the bulk value (2.1 cm^{-1}) as shown in Table I.” to “The magnetic field dependence of the MP modes in 4L flakes is presented in Fig. 4c. The field-induced shift of all four MP modes can be resolved as marked by the arrows. The corresponding peak positions can be well described with the selective strong coupling model, as shown in Fig 4d. The extracted uncoupled magnon frequency of 4L flakes is 113.2 cm^{-1} , exhibiting a slight redshift to its bulk value, while the magnon-phonon coupling strength remains strong reaching 2.0 cm^{-1} , as listed in Table I.”
8. Line 198, added “both bulk and”
9. In Fig.4, updated new Raman spectra of the 4L FePSe₃, removing the colormap for the 3L sample, added 4L FePSe₃ flake peak position and fit as Fig.4d.
10. In Fig.5, update Fig.5d and 5e with corrected Raman calculation results.
11. Line 249-257, added a paragraph in the main text describing magnon-induced chiral phonon. “We note that without magnon-phonon coupling, the two (nearly) degenerate phonon modes can equally well be considered as either a superposition of two linearly polarized phonons or two circularly polarized phonons. They would remain nearly unchanged under external fields due to the negligibly small orbital phonon magnetic moments. It is the magnon-phonon coupling that effectively lifts the degeneracy between the two circularly polarized phonons with opposite angular momentum under magnetic fields. This happens because of the selective form of the coupling, where the right-handed (left-handed) magnon couples to the right-handed (left-handed) phonon. The magnon-phonon coupling is thus what induces the phonon chirality.”

Reviewers' Comments:

Reviewer #1:

Remarks to the Author:

The authors have adequately revised the manuscript according to the reviewers' comments. I recommend the manuscript for publication.

Reviewer #2:

Remarks to the Author:

I read the authors responses. I am satisfied with their replies and manuscript revisions. I recommend acceptance.

Reviewer #3:

Remarks to the Author:

The authors have adequately answered my questions and comments. I find the revised manuscript to be ready for publication in Nature Communications.

Response to the second-round reviews for "Chirality selective magnon-phonon hybridization and magnon-induced chiral phonons in a layered zigzag antiferromagnet"

We are grateful for the referees' careful review of our manuscript and for their comments, questions, and suggestions, which we address below and in the revised manuscript. Referee remarks are in *blue italic font*.

Report of Referee #1

The authors have adequately revised the manuscript according to the reviewers' comments. I recommend the manuscript for publication.

Report of Referee #2

I read the authors responses. I am satisfied with their replies and manuscript revisions. I recommend acceptance.

Report of Referee #3

The authors have adequately answered my questions and comments. I find the revised manuscript to be ready for publication in Nature Communications.

Our response:

We thank the referees for recommending the publication of our research in Nature Communications. We appreciate all three referees for their efforts in reviewing the manuscript and providing valuable comments and suggestions, which substantially helped us improve the paper.